# Comparing Comparisons: Informative and Easy Human Feedback with Distinguishability Queries

**Xuening Feng** [1]   **Zhaohui Jiang** [1]   **Timo Kaufmann** [2 3]   **Eyke Hüllermeier** [2 3 4]   **Paul Weng** [5]   **Yifei Zhu** [1]

## Abstract

Learning human objectives from preference feedback has significantly advanced reinforcement learning (RL) in domains where objectives are hard to formalize. However, traditional methods based on pairwise trajectory comparisons face notable challenges, including the difficulty in comparing trajectories with subtle differences and the limitation of conveying only ordinal information, limiting direct inference of preference strength. In this paper, we introduce a novel *distinguishability query*, enabling humans to express preference strength by comparing two pairs of trajectories. Labelers first indicate which of two pairs is easier to distinguish, then provide preference feedback only on the easier pair. Our proposed query type directly captures preference strength and is expected to reduce the cognitive load on the labeler. We further connect this query to cardinal utility and difference relations and develop an efficient query selection scheme to achieve a better trade-off between query informativeness and easiness. Experimental results demonstrate the potential of our method for faster, data-efficient learning and improved user-friendliness in RLHF benchmarks, particularly in classical control settings where preference strength is critical for expected utility maximization.

## 1. Introduction

Learning human objectives from preference feedback has enabled reinforcement learning (RL) to tackle domains with hard-to-formalize objectives, from fine-tuning language

[1]Shanghai Jiao Tong University, Shanghai, China [2]Institute for Informatics, LMU Munich, Munich, Germany [3]Munich Center of Machine Learning (MCML), Munich, Germany [4]German Research Center for Artificial Intelligence (DFKI) [5]Digital Innovation Research Center, Duke Kunshan University, Kunshan, China. Correspondence to: Paul Weng <paul.weng@dukekunshan.edu.cn>.

*Proceedings of the $42^{nd}$ International Conference on Machine Learning*, Vancouver, Canada. PMLR 267, 2025. Copyright 2025 by the author(s).

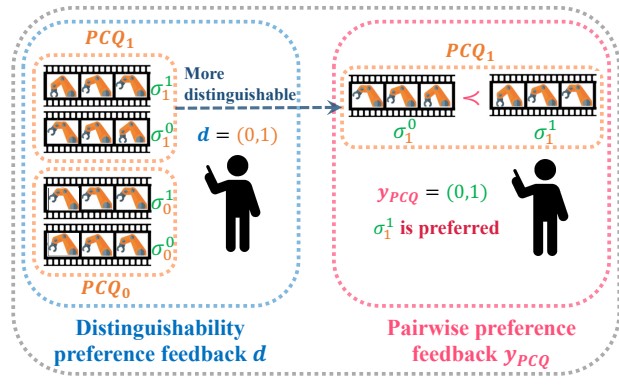

*Figure 1.* An illustration of the distinguishability query. PCQ refers to the usual Pairwise Comparison Query for *a pair* of segments $(\sigma^0, \sigma^1)$, and its feedback is $y_{\text{PCQ}}$, indicating the preferred segment. Our proposed distinguishability query, DQ, consists of *two* candidate PCQs. Its feedback $y_{\text{DQ}} = (d, y_{\text{PCQ}})$ includes the choice of the more distinguishable PCQ, $d$, and its corresponding label, $y_{\text{PCQ}}$.

models (OpenAI, 2022; Ouyang et al., 2022) to training robots for complex tasks (Christiano et al., 2017). The standard method asks human evaluators to compare pairs of trajectories, using these comparisons to learn a reward function that guides RL agent training. While this method is widely used and has shown impressive performance in simulated environments (Christiano et al., 2017; Lee et al., 2021b; Liang et al., 2022; Park et al., 2022; Hu et al., 2024; Verma & Metcalf, 2024; Dong et al., 2024), it faces two key limitations: First, humans often struggle to compare trajectories, particularly when the differences are subtle, leading to inefficient queries and poor user experience. Second, pairwise comparisons provide only ordinal information, leaving preference strength to be learned implicitly. This limitation is particularly problematic for reinforcement learning, which fundamentally requires cardinal utilities for expected utility maximization – the core principle guiding the agent's decisions under uncertainty.

In this paper, we propose a novel query type called the *distinguishability query* (illustrated in Figure 1), which addresses

these two critical limitations. In a distinguishability query, the human first indicates which of two trajectory comparisons is *easier to distinguish* and then provides preference feedback on *only the selected pair* as usual in preference-based RLHF methods. While traditional preference choice for a single pair of segments only conveys ordinal information, the additional choice of which comparison to evaluate allows us to infer preference strength, assuming that humans prefer to provide feedback on the more distinguishable pair. This approach enables labelers to focus on easier comparisons and directly facilitates the learning of preference strength. Our proposed method, named DistQ, integrates distinguishability queries with an effective query selection scheme and a specialized learning objective.

Concretely, our contributions are as follows:

1. We propose a new type of query, the distinguishability query, in the field of RLHF (see Section 4.1);

2. We establish connections between distinguishability queries and cardinal utility and distance relations studied in related disciplines (see Section 2);

3. We design a query selection scheme for distinguishability queries that aims to achieve a better trade-off between query informativeness and ease of answering (see Section 4.2);

4. We propose a specific learning objective to improve reward learning by coupling cardinal and ordinal information (see Section 4.3);

5. We empirically demonstrate, on classic control tasks with a synthetic oracle, that our method can achieve competitive performance in a more user-friendly manner compared to standard pairwise comparison methods (see Section 5).

## 2. Related Work

Our work overlaps with three key areas: decision theory, RLHF approaches using pairwise comparisons, and approaches that aim to reduce the burden on the human labeler. We briefly review each of these areas below.

**Preference Strength**    Observed choices, which form the foundation of RLHF (Jeon et al., 2020), convey only *ordinal* preferences, expressing information about rank but not about distance. With cardinal utilities, not only the order of utility values but also their differences are meaningful – capturing preference strength, which is not expressed in purely ordinal utilities. Reinforcement learning fundamentally requires such cardinal utilities for expected utility maximization (Von Neumann & Morgenstern, 1947). Cardinal utilities have been extensively studied in economics,

psychology, and decision theory (Suppes & Winet, 1955; Krantz et al., 2006; Jansen et al., 2018). Prior work has shown that preferences elicited in RLHF, when noisy and under certain assumptions, can be used to infer cardinal utilities (Chan et al., 2021; Xu et al., 2020). This aligns with the empirical success of RLHF methods based on pairwise comparisons (Lee et al., 2021b; Liang et al., 2022; Park et al., 2022; Hu et al., 2024; Dong et al., 2024). However, these assumptions about utility-dependent noise are strong and may not hold in practice. Moreover, since each comparison is typically observed only once, models must rely on generalization across similar comparisons to infer preference strength. This motivates us to explore more direct ways to elicit cardinal utilities.

**Distance Relations**    Cardinal utilities can be represented either as a real-valued function (unique up to positive affine transformations) or as a relation on pairs of outcomes. The latter formalism aligns closely with our proposed query type, where human labelers distinguish between two pairs of outcomes. Formally, human labeler preferences can be modeled using two relations $R$ and $D$ (Suppes & Winet, 1955; Jansen et al., 2018), where $R$ represents a preference relation and $D$ a difference relation. If a pair of pairs satisfies $((a, b), (c, d)) \in D$, then exchanging $b$ with $a$ is at least as desirable as exchanging $d$ with $c$ – meaning $a$ is more strongly preferred over $b$ than $c$ is over $d$. Several works establish axioms that determine a utility function from such a relation, unique up to positive affine transformations (Alt, 1936; Suppes & Winet, 1955; Köbberling, 2006). Notable among these axioms are completeness and transitivity. When completeness is not satisfied, the relation only determines a set of compatible utility functions (Pivato, 2013).

**Eliciting Preference Strength**    Eliciting distinguishability for cardinal utility in RLHF remains underexplored. While Jansen et al. (2022) explore methods like *time elicitation* and *label elicitation* (the latter demanding absolute ratings, posing challenges (Yannakakis & Martínez, 2015) and burdening labelers), these are suited for finite outcome spaces with dense observations. DistQ instead employs *relative* distinguishability queries and integrates neural network utility learning with probabilistic modeling and informativeness-based query selection for generalization across vast trajectory spaces with learnable structure. Other methods infer cardinal strength from human response times (RTs), assuming RTs reflect preference strength through explicit cognitive modeling using the Drift Diffusion Model (DDM) (Ratcliff, 1978; Ratcliff & McKoon, 2008) or by leveraging RTs as predictive features. Shvartsman et al. (2024) use Gaussian Processes with either DDM approximations or RTs as stacking features, while Li et al. (2024) use EZ-DDMs (Wagenmakers et al., 2007) in linear bandits (assuming lin-

ear utility). DistQ's explicit, second-order query operates differently: it asks humans to identify which of the two *potential* choices is easier, leveraging their meta-cognitive assessment of relative difficulty to allow an answer and provide strength signal while avoiding fully resolving the harder underlying comparison. This query type aims for greater robustness by avoiding direct reliance on absolute RTs and strong accompanying assumptions (e.g., DDM, linear utility) – problematic in complex RLHF settings – and its distinguishability signal readily extends standard loss functions.

**Reducing Burden of Human Labelers**   A major limitation of pairwise comparisons is their burden on human labelers when the compared behaviors are similar or when neither option is clearly preferable. Prior work has addressed this through multiple strategies: (1) pre-training, either in an unsupervised manner (Lee et al., 2021b) or using demonstrations (Ibarz et al., 2018; Palan et al., 2019; Bıyık et al., 2022), (2) allowing labelers to abstain from answering queries (Lee et al., 2021a), and (3) query selection strategies that aim to select queries that are easier for the human labeler to answer (Bıyık et al., 2019). The first two strategies are complementary to our approach. Regarding the third, which our approach falls into, Bıyık et al. (2019) propose to use information gain to select queries that are informative and easy to answer, implicitly prioritizing queries the human will be able to answer and thus lead to the largest gain of information. While such an approach could be extended to distinguishability queries, we instead opted for a query selection scheme based on ensemble disagreement and prediction entropy for computational efficiency. In addition to this selection scheme, our distinguishability queries allow the labeler to actively choose easier queries themselves.

## 3. Preliminaries

**Reinforcement Learning**   We consider a reinforcement learning (RL) setting characterized by a Markov Decision Process (MDP). The MDP is defined by $\langle \mathcal{S}, \mathcal{A}, \mathcal{P}, r, \gamma \rangle$, where $\mathcal{S}$ and $\mathcal{A}$ denote state and action spaces, while $\mathcal{P}(s'|s, a)$, $r(s, a)$, and $\gamma \in (0, 1]$ represent the transition function, the reward function, and the discount factor. At each timestep $t$, the agent receives state $s_t \in \mathcal{S}$, takes action $a_t \in \mathcal{A}$, receives reward $r(s_t, a_t)$, and transitions to $s_{t+1} \sim \mathcal{P}(s_{t+1}|s_t, a_t)$. The return $\mathcal{R}_t = \sum_{k=0}^{\infty} \gamma^k r(s_{t+k}, a_{t+k})$ is defined as the discounted cumulative sum of rewards from timestep $t$, which is to be optimized.

**Reinforcement Learning from Human Feedback**   RLHF is a framework that aims to learn optimal agent behavior from human feedback (Christiano et al., 2017; Kaufmann et al., 2025). In this paper, we focus on inferring an unknown

reward function $r(s, a)$ used to train policy $\pi(a|s)$ (Lee et al., 2021b;a). The agent learns an approximate reward function $\hat{r}_\psi(s, a)$ from human preferences, implemented as an ensemble of $N$ neural networks parameterized by $\psi = (\psi_1, \ldots, \psi_N)$. The policy $\pi_\phi$ and reward $\hat{r}_\psi$ are updated by alternating:

- *Step 1 (agent learning)*: The agent interacts with the environment using $\pi_\phi$ to collect trajectories and updates the policy via RL, maximizing expected return under the current $\hat{r}_\psi$.

- *Step 2 (reward learning)*: Preference queries are generated from collected trajectories and used to update the reward model $\hat{r}_\psi$ based on human feedback.

While any RL algorithm could be used in Step 1, we use the Soft Actor-Critic (SAC) algorithm (Haarnoja et al., 2018) in our experiments, following PEBBLE (Lee et al., 2021a).

**Pairwise Comparison Queries in RLHF**   In RLHF, human feedback is typically collected through pairwise comparison queries (PCQ) (Christiano et al., 2017; Lee et al., 2021b). Given trajectory segments $\sigma^0$ and $\sigma^1$, represented by a sequence of states and actions, a PCQ can be denoted as $(\sigma^0, \sigma^1)$. The oracle (e.g., human labeler) gives preference through feedback $y_{\text{PCQ}} \in \{(1, 0), (0, 1)\}$, where $(1, 0)$ indicates segment $\sigma^0$ is preferred over $\sigma^1$ and $(0, 1)$ indicates the opposite. These query-feedback triples $(\sigma^0, \sigma^1, y_{\text{PCQ}})$ are stored in dataset $\mathcal{D}_{\text{PCQ}}$.

Preferences are linked to the reward function by means of a Bradley-Terry model (Bradley & Terry, 1952), which assumes latent utilities $p^j$ govern pairwise preferences through

$$\mathbb{P}[\sigma^1 \succ \sigma^0] = \frac{\exp(p^1)}{\exp(p^0) + \exp(p^1)}.$$

In the context of RLHF, where the utility of a trajectory segment is defined as its return, the predicted probability of segment $\sigma^1$ being preferred over $\sigma^0$ is

$$\mathbb{P}_\psi[\sigma^1 \succ \sigma^0] = \frac{\exp \sum_t \gamma^t \hat{r}_\psi(s_t^1, a_t^1)}{\sum_{j \in \{0,1\}} \exp \sum_t \gamma^t \hat{r}_\psi(s_t^j, a_t^j)}, \quad (1)$$

where $\hat{r}_\psi(s_t^j, a_t^j)$ for $j \in \{0, 1\}$ is the average output of the $N$ reward networks $\hat{r}_{\psi_i}$ for $i \in \{1, \ldots, N\}$.

Given dataset $\mathcal{D}_{\text{PCQ}}$ and corresponding predictions from Equation (1), reward learning is formulated as a classification problem (Christiano et al., 2017). The reward model $\hat{r}_\psi$ can be learned by minimizing the cross-entropy loss

$$\mathcal{L}_{\text{PCQ}}^{\text{Rew}} = -\mathbb{E}_{\mathcal{D}_{\text{PCQ}}}\Big[ \sum_{j \in \{0,1\}} y_{\text{PCQ}}^j \log \mathbb{P}_\psi\big[\sigma^j \succ \sigma^{1-j}\big]\Big], \quad (2)$$

where $y_{\text{PCQ}} = (y_{\text{PCQ}}^0, y_{\text{PCQ}}^1)$ is the feedback and $\mathbb{E}_{\mathcal{D}^{\text{PCQ}}}$ is the (empirical) expectation for $(\sigma^0, \sigma^1, y_{\text{PCQ}}) \sim \mathcal{D}_{\text{PCQ}}$.

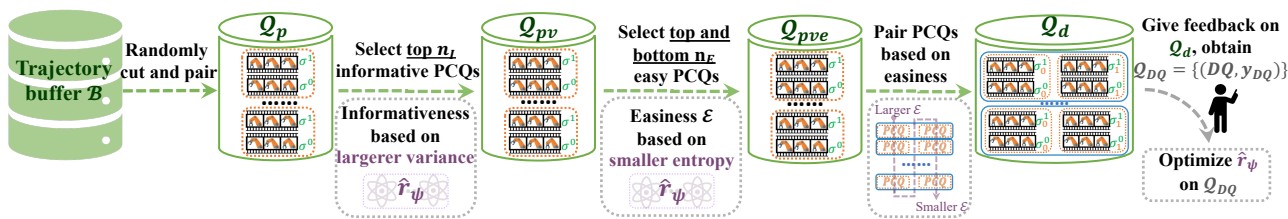

*Figure 2.* DistQ's query selection scheme for distinguishability queries. The process consists of multiple filtering steps: First, from a set of randomly generated segment pairs $\mathcal{Q}_p$, we select a subset $\mathcal{Q}_{pv}$ with higher informativeness $\mathcal{I}$. Then, we filter a subset $\mathcal{Q}_{pve}$ based on easiness $\mathcal{E}$ from $\mathcal{Q}_{pv}$. Both $\mathcal{I}$ and $\mathcal{E}$ are calculated from the current reward model $\hat{r}_\psi$. Finally, we construct $\mathcal{Q}_d$ by pairing the PCQs in $\mathcal{Q}_{pve}$, matching the top $i$-th easiest PCQ to the bottom $(i + n_E)$-th one. We obtain $\mathcal{Q}_{DQ}$ with the labelers' feedback on $\mathcal{Q}_d$, which can then be utilized by our training objective for $\hat{r}_\psi$, integrating information from distinguishability and preference feedback.

## 4. Method

We present DistQ, a novel RLHF approach that addresses limitations of traditional pairwise comparisons through distinguishability queries. Our method aims to pose queries that are both informative for distance-aware reward learning and easier for human labelers to answer, enhancing both user experience and query efficiency. The approach consists of three key components: (1) the distinguishability query structure that enables humans to express preference strength by comparing pairs of trajectory comparisons, (2) an efficient query selection method considering both informativeness and easiness, and (3) a specialized learning objective that leverages the richer feedback from distinguishability queries to improve reward learning.

The remainder of this section details these components. In Section 4.1, we introduce the structure and theoretical foundations of distinguishability queries. Section 4.2 presents our efficient query selection scheme designed specifically for these queries. Finally, Section 4.3 develops a specialized training objective that integrates both distinguishability and preference feedback. Together, these components form a cohesive system with the aim of creating user-friendly, informative queries while enriching the reward learning process, as illustrated in Figure 2.

### 4.1. Distinguishability Query

Intuitively, in order to avoid posing unanswerable PCQs and to learn about preference strength, we propose providing the labeler with two such queries together as one distinguishability query. We let the labeler select the more distinguishable one that is easier to answer and then provide preference feedback for the chosen pairwise query. This approach effectively combines a query about ordinal preferences with one about preference strength, while simultaneously reducing the burden on the human labeler.

Recall that in Section 3, the pairwise comparison query is denoted as PCQ $= (\sigma^0, \sigma^1)$ and the corresponding pref-

erence feedback as $y_{\text{PCQ}} \in \{(1, 0), (0, 1)\}$. We represent the **distinguishability query** as DQ $= (\text{PCQ}_0, \text{PCQ}_1) = ((\sigma_0^0, \sigma_0^1), (\sigma_1^0, \sigma_1^1))$. The feedback to a distinguishability query $y_{\text{DQ}} = (d, y_{\text{PCQ}})$ consists of two components: the **distinguishability preference feedback** $d \in \{(1, 0), (0, 1)\}$ indicating which pairwise comparison query is more distinguishable, and the **pairwise preference feedback** $y_{\text{PCQ}}$ to the selected more distinguishable pairwise comparison query. Such a query and corresponding feedback is represented by $(\text{DQ}, y_{\text{DQ}})$ and stored in a dataset $\mathcal{D}_{\text{DQ}}$. See Figure 1 for an illustration.

We define the **distinguishability measure** $\mathcal{M}$ for a pairwise comparison query $(\sigma^0, \sigma^1)$ as

$$\mathcal{M}(\sigma^0, \sigma^1) = \Big| \sum_t \gamma^t r(s_t^1, a_t^1) - \sum_t \gamma^t r(s_t^0, a_t^0) \Big|. \quad (3)$$

Larger values indicate stronger distinguishability. Note that $\mathcal{M}$ is symmetric. Given a learned reward model $\hat{r}_\psi$, distinguishability $\mathcal{M}(\sigma^0, \sigma^1)$ can be approximated by

$$\hat{\mathcal{M}}_\psi(\sigma^0, \sigma^1) = \Big| \sum_t \gamma^t \hat{r}_\psi(s_t^1, a_t^1) - \sum_t \gamma^t \hat{r}_\psi(s_t^0, a_t^0) \Big|. \quad (4)$$

### 4.2. Selecting Informative and Easy-to-Answer Distinguishability Queries

Given this newly proposed type of query, we design a method (see Figure 2) to select distinguishability queries that are informative and also easy to answer for the labeler. Overall, we begin with selecting desirable pairwise comparison queries and pair the selected ones into distinguishability queries.

Broadly speaking, we aim to select queries on which feedback from the labeler could reduce our current predictive uncertainty while also ensuring that the labeler can easily provide feedback. This aligns with the concepts of epistemic and aleatoric uncertainty (Hüllermeier & Waegeman, 2021): While the latter refers to inherent uncertainty due to randomness in the data-generating process (in our case the

labeler's responses), the former is caused by the learner's limited knowledge of this process. Thus, while aleatoric uncertainty is irreducible, epistemic uncertainty can, in principle, be reduced through additional (training) information and therefore is a natural target for active learning and query construction (Nguyen et al., 2022). Since we assume the labeler gives stochastic feedback according to a Bradley-Terry model, aleatoric uncertainty is high when the utility difference is small and low when the utility difference is large.

Our selection scheme, therefore, favors queries that are informative and hence epistemically uncertain, and meanwhile easy to answer by the labeler. Following previous work (Christiano et al., 2017; Lee et al., 2021b), we measure the informativeness using the variance of the prediction of the reward networks $\mathbb{P}_{\psi_i}$'s. A larger variance means larger disagreement between the reward networks and, thus, higher epistemic uncertainty. For query easiness, we measure it using the average entropy of the Bernoulli distributions $\mathbb{P}_{\psi_i}$'s. Indeed, a large entropy implies that the return difference of two segments is small, which means that they are hard to distinguish. Next, we will explain our method step by step in detail.

**Informativeness Based on Variance**    As Figure 2 shows, given a trajectory buffer $\mathcal{B}$, we first randomly sample segments from trajectories and pair them to obtain a set $\mathcal{Q}_p$ of candidate PCQs. With the current reward model $\hat{r}_\psi$, we can compute the predicted pairwise preference probability in Equation (1) for each PCQ $(\sigma^0, \sigma^1) \in \mathcal{Q}_p$.

We define the informativeness $\mathcal{I}(\sigma^0, \sigma^1)$ of a PCQ $(\sigma^0, \sigma^1)$ as the variance of reward model prediction, which is

$$\mathcal{I}(\sigma^0, \sigma^1) = \mathbb{V}(\sigma^0, \sigma^1) = \sqrt{\frac{1}{N} \sum_{i=1}^{N} \left( P_{\psi_i}^1 - P_\psi^1 \right)^2}, \quad (5)$$

where $P_{\psi_i}^1 = \mathbb{P}_{\psi_i}[\sigma^1 \succ \sigma^0]$ is the prediction solely from neural network $\hat{r}_{\psi_i}$ and $P_\psi^1 = \mathbb{P}_\psi[\sigma^1 \succ \sigma^0]$ is the average prediction from the ensemble reward model. Queries with higher variance indicate higher epistemic uncertainty of the current reward model, thus providing more information for reward learning. In this step, PCQs with top $n_I$ informativeness are finally selected from set $\mathcal{Q}_p$ for later steps.

**Easiness Based on Entropy**    Let $\mathcal{Q}_{pv}$ be the set of the $n_I$ informative queries obtained in the last step. We then define the easiness $\mathcal{E}(\sigma^0, \sigma^1)$ of a PCQ $(\sigma^0, \sigma^1)$ as the negative entropy of reward model prediction, which is

$$\mathcal{E}(\sigma^0, \sigma^1) = -\mathcal{H}(\hat{r}_\psi) = \sum_{j=0}^{1} P_\psi^j \log P_\psi^j, \quad (6)$$

where $P_\psi^j = \mathbb{P}_\psi[\sigma^j \succ \sigma^{1-j}]$ for $j \in \{0, 1\}$ represents the average prediction from the ensemble reward model.

It is worthwhile to mention that although higher entropy values may also correspond to more uncertain predictions, and the entropy criterion has been used for pairwise comparison query selection (Lee et al., 2021b), relying only on the highest entropy can result in queries that are nearest to the decision boundary and thus really hard for the labeler to answer. By selecting queries with lower entropy among those selected with the largest variance, we aim to select the queries that are as easy to answer despite the epistemic uncertainty involved.

**Forming Distinguishability Queries**    After sorting the $n_I$ queries in $\mathcal{Q}_{pv}$ in decreasing order with respect to their easiness $\mathcal{E}(\sigma^0, \sigma^1)$, the top $n_E$ and bottom $n_E$ easiest queries are selected to form the set $\mathcal{Q}_{pve}$. Then the top $i$-th easiest query is paired with the bottom $(i + n_E)$-th easiest one to compose a distinguishability query. We apply this simple method so that easy queries are paired with less easy queries while still considering only the most informative queries overall.

### 4.3. Training with Distinguishability Queries

To learn from distinguishability feedback, we assume that the labeler chooses between $\text{PCQ}_0 = (\sigma_0^0, \sigma_0^1)$ and $\text{PCQ}_1 = (\sigma_1^0, \sigma_1^1)$ according to a Bradley-Terry model. Using reward model $\hat{r}_\psi$, the labeler's response can be predicted by:

$$\widetilde{P}_\psi[(\sigma_1^0, \sigma_1^1) \succ (\sigma_0^0, \sigma_0^1)] = \frac{\exp \hat{\mathcal{M}}_\psi(\sigma_1^0, \sigma_1^1)}{\sum_{h \in \{0,1\}} \exp \hat{\mathcal{M}}_\psi(\sigma_h^0, \sigma_h^1)}, \quad (7)$$

where $(\sigma_1^0, \sigma_1^1) \succ (\sigma_0^0, \sigma_0^1)$ means that $\text{PCQ}_1$ is more distinguishable than $\text{PCQ}_0$.

Under this assumption, we can then train the reward model with feedback $d$ by formulating the distinguishability preference prediction also as a supervised classification problem. Similarly to Equation (2), we define the cross-entropy loss for the distinguishability preference feedback $d$ as

$$\mathcal{L}_d^{\text{Rew}} = -\mathbb{E}_{\mathcal{D}_{\text{DQ}}} \Big[ d_0 \log \widetilde{P}_\psi \big[ (\sigma_0^0, \sigma_0^1) \succ (\sigma_1^0, \sigma_1^1) \big]$$
$$+ d_1 \log \widetilde{P}_\psi \big[ (\sigma_1^0, \sigma_1^1) \succ (\sigma_0^0, \sigma_0^1) \big] \Big], \quad (8)$$

where $\mathbb{E}_{\mathcal{D}_{\text{DQ}}}$ represents the (empirical) expectation for $((\sigma_0^0, \sigma_0^1), (\sigma_1^0, \sigma_1^1), y_{\text{DQ}}) \sim \mathcal{D}_{\text{DQ}}$. Recall that distinguishability feedback $y_{\text{DQ}} = (d, y_{\text{PCQ}})$ consists of both distinguishability preference feedback $d$ and pairwise preference feedback $y_{\text{PCQ}}$. The latter feedback $y_{\text{PCQ}}$ can also be exploited for reward training by writing loss $\mathcal{L}_{\text{PCQ}}^{\text{Rew}}$ in Equation (2) on the pairwise comparison query selected by $d$.

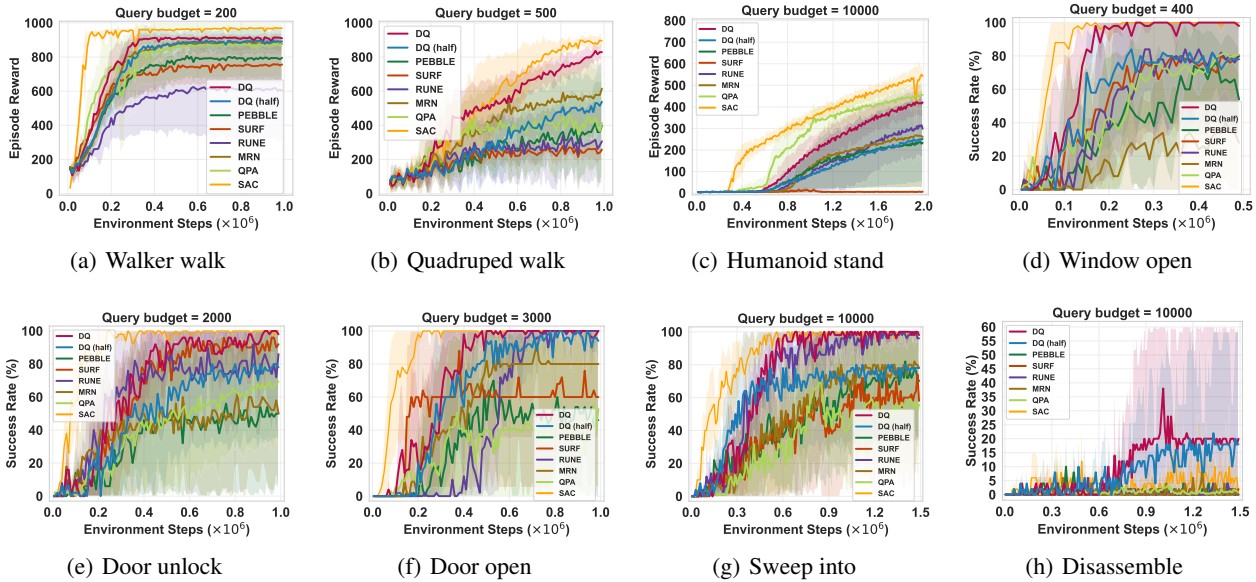

*Figure 3.* Performance comparison of DistQ against baselines on classic control tasks. Results show episode reward for DMControl locomotion tasks (Figures 3(a) to 3(c)) and success rate for Meta-World manipulation tasks (Figures 3(d) to 3(h)). To account for DistQ's different query structure, we show 'DQ' (same query budget as baselines) and 'DQ (half)' (half query budget) to bracket the human effort for comparison. The solid curves and shaded regions represent the mean and standard deviation, respectively, across five runs.

Finally, we update the reward model $\hat{r}_\psi$ by minimizing the linear combination of $\mathcal{L}_d^{\text{Rew}}$ and $\mathcal{L}_{\text{PCQ}}^{\text{Rew}}$ as

$$\mathcal{L}_{\text{DQ}}^{\text{Rew}} = \lambda_d \mathcal{L}_d^{\text{Rew}} + \lambda_p \mathcal{L}_{\text{PCQ}}^{\text{Rew}}, \qquad (9)$$

where hyperparameters $\lambda_d$ and $\lambda_p$ denote the weight for $\mathcal{L}_d^{\text{Rew}}$ and $\mathcal{L}_{\text{PCQ}}^{\text{Rew}}$, respectively.

Our proposition, DistQ, corresponds to enhancing Step 2 (reward learning) described in Section 3. For simplicity, we do not change Step 1 (agent learning). The pseudo-code for the general procedure can be found in Algorithm 1 in Appendix C.

## 5. Experiments

In this section, we conduct experiments to investigate the following questions: (1) How do the proposed distinguishability query and corresponding query selection method help with performance and query efficiency compared with state-of-the-art (SOTA) RLHF methods that only use pairwise comparison queries? (2) Are the pairwise comparison queries selected by our method easier to answer compared to those selected by the baseline methods? (3) How does each proposed technique contribute to the overall design?

### 5.1. Experimental Setup

**Tasks** Similar to prior works (Lee et al., 2021b;a; Park et al., 2022; Liang et al., 2022; Liu et al., 2022; Hu et al., 2024), we consider a series of locomotion tasks from the

DeepMind Control Suite (DMControl) (Tassa et al., 2018) and robotic manipulation tasks from the Meta-World benchmark (Yu et al., 2019).

To quantitatively evaluate the performance of the involved RLHF methods, we follow a general setting where the agent has no access to the ground truth reward from the environment but can only receive synthetic feedback based on the ground truth reward from a scripted labeler (Lee et al., 2021b). Given the feedback, the agent learns to solve the corresponding task guided by the underlying reward function. The performance is then measured as the true average return for locomotion tasks and the success rate for manipulation tasks. We report the mean and standard deviation across five runs for all experiments.

**Baselines** For comparison, we adopt a variety of SOTA methods in the field of RLHF, including PEBBLE (Lee et al., 2021b), SURF (Park et al., 2022), RUNE (Liang et al., 2022), MRN (Liu et al., 2022), and QPA (Hu et al., 2024). All of these baseline methods use pairwise comparison queries. Regarding query selection, PEBBLE adopts an entropy-based method (i.e., smaller $\mathcal{E}$), while SURF, RUNE, and MRN adopt a variance-based method (i.e., larger $\mathcal{I}$). QPA selects queries randomly from a buffer of near-on-policy trajectories. All baselines are evaluated with the original settings listed in their paper. More details are provided in Appendix A. What is more, considering that all these methods employ SAC for agent learning, we also mea-

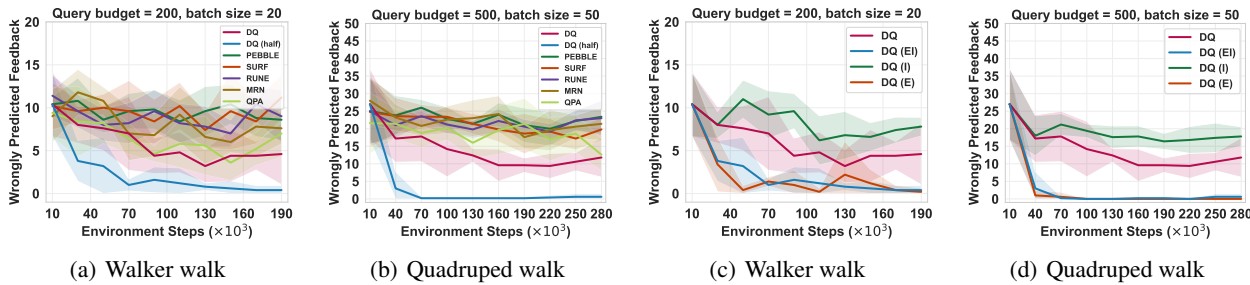

(a) Walker walk      (b) Quadruped walk      (c) Walker walk      (d) Quadruped walk

*Figure 4.* Number of incorrectly predicted pairwise comparison feedback across training iterations on locomotion tasks. Lower values indicate queries that are easier for the labeler to answer, as the reward model's predictions better align with the labeler's preferences. Note that this figure includes results of ablation studies, as detailed in Section 5.5. The solid curves and shaded regions represent the mean and standard deviation, respectively, across five runs.

sure the performance of SAC using the ground truth reward function as an upper bound of performance.

**Implementation** We implement the distinguishability query and the query selection method on top of the widely adopted method PEBBLE (Lee et al., 2021b). We also adopt the same generic hyperparameter settings as PEBBLE without any tuning for a fair comparison. This implementation is then evaluated and compared with all baselines. We argue that the proposed new query and corresponding query selection method can be implemented on top of any RLHF method utilizing pairwise comparison queries. See Appendix B for more details on the implementation.

### 5.2. Benchmark Tasks with Unobserved Rewards

Figure 3 shows the learning curves of DistQ and the five baselines on three locomotion tasks (*Walker walk*, *Quadruped walk*, and *Humanoid stand*) and five robotic manipulation tasks (*Window open*, *Door unlock*, *Door open*, *Sweep into*, and *Disassemble*). All baseline methods use a budget of pairwise comparison queries indicated by "Query budget" in each subfigure.

Note that for DistQ, although a distinguishability query is composed of two pairwise comparison queries, we only ask **the more distinguishable pairwise comparison query** to the labeler. Therefore, we show the results of DistQ using both full budget (i.e., the DQ curve) and half budget (i.e., the DQ (half) curve), aiming for a fair comparison that lies somewhere in between these cases. In Figure 3(a), for example, the curves of all baseline methods are obtained by asking 200 pairwise comparison queries, while DQ (half) uses only 100 distinguishability queries. It is indeed difficult to set a precise budget for DistQ to enable a completely fair comparison. However, by presenting the results in this manner, we believe that the full and half budget settings provide a performance range of DistQ, which can serve as a meaningful comparison with baselines.

**Locomotion Tasks from DMControl** As shown in Figures 3(a) to 3(c), DistQ with a full budget largely outperforms the baseline methods. This aligns with our expectations, as distinguishability queries naturally provide richer information for reward learning, leading to superior reward models for guiding the agent. The primary exception is the challenging Humanoid stand task, where QPA outperforms DistQ with full budget slightly. Notably, Humanoid stand has not been addressed by most other baselines, prompting us to adopt QPA's hyperparameter settings of the total query budget and the query batch size in one feedback session for all methods on this specific task for a fair comparison. Additionally, even with only half the budget, DistQ demonstrates competitive performance compared with most baselines (excluding QPA). Across the other tasks, DQ (half) generally surpasses most baselines, with MRN on Quadruped walk being the other exception. The ability of DQ (half) to match or outperform many of the baselines despite the reduced query budget demonstrates distinguishability queries are efficient while providing rich information and allowing for selective answering of easier queries.

DistQ shares PEBBLE as the common foundation with the baselines (see Appendix A). As these baselines focus mainly on different aspects to improve query efficiency and performance, such as exploration, augmentation of unlabeled data, and new training procedures for the agent, which are orthogonal to DistQ, we expect that combining DistQ with each of the baselines could further improve performance, which would be worthwhile to explore for future work.

**Robotic Manipulation Tasks from Meta-World** The results in Figures 3(d) to 3(h) showcase similar phenomena to the locomotion results. DistQ with full budget still outperforms all baselines and even converges to the same performance as SAC (yellow) or outperforms SAC. The version with half budget also exhibits better or similar performance compared with baselines with the exception of RUNE on Sweep into and SURF on Door unlock.

These results effectively demonstrate that the proposed DistQ can generally improve both performance and query efficiency. We present the results numerically in Appendix D.1 for a clearer comparison.

## 5.3. Query Easiness

Typically, in reward learning, a batch of pairwise comparison queries is selected and presented to a human labeler during each feedback session. We consider queries to likely be difficult to answer when the reward model's prediction for a pairwise comparison is inconsistent with the labeler's actual response. The model, trained on prior feedback, should discern clear preferences. An inconsistency therefore suggests the query involved subtle differences, conflicting attributes, or unmastered nuances. These cognitively demanding queries align with the concept of "wrong answers" that Bıyık et al. (2019) describe, as they are more prone to human error and deliberation.

To investigate whether we ask easier pairwise comparison queries than baselines, we show curves of incorrectly predicted feedback to pairwise comparison queries along the training process of various methods in Figures 4(a) and 4(b). Note that here DistQ with half budget (blue) also adopts half of the query batch size. We see that among all methods, DistQ with full budget (pink) always makes the fewest mispredictions given the same number of pairwise comparison queries allowed to all baselines, which indicates that DistQ can really ask easier-to-answer queries and thus reduce the labeler's effort.

## 5.4. User Study

To provide a more direct validation of DistQ's effectiveness and user-friendliness, we conduct a user study involving a real human labeler. This study compares DistQ against its backbone method, PEBBLE. For this evaluation, we design a novel task within the Quadruped environment: the agent is intended to learn to stand and then wave its right hind leg. The human participant provides feedback on the agent's behavior based on short video demonstrations, guiding the learning process for both methods.

The agent's learned behavior is evaluated at the end of the training period for both DistQ and PEBBLE. The results highlight a significant performance difference: the agent trained with DistQ successfully performs the desired behavior in 100% of 10 evaluation rounds. In contrast, the agent trained with PEBBLE fails to perform the behavior in any round (0% success rate), often struggling even to achieve a stable standing posture. Furthermore, qualitative feedback from the participant indicates that providing feedback for DistQ is perceived as noticeably easier compared to PEBBLE, suggesting a reduced cognitive burden on the human with DistQ. More comprehensive details of the user study

settings and results can be found in Appendix D.3.

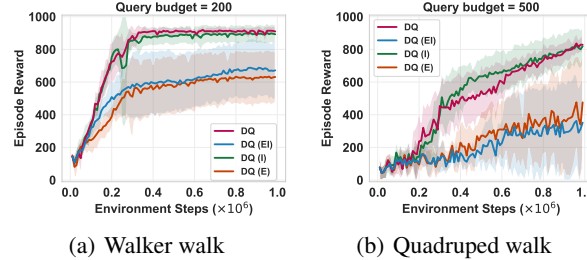

(a) Walker walk      (b) Quadruped walk

*Figure 5.* Ablation study evaluating different **query selection strategies**. Learning curves on locomotion tasks illustrate the comparison between our full method (DQ) and variants prioritizing easiness then informativeness (EI), only informativeness (I), or only easiness (E). The solid curves and shaded regions represent the mean and standard deviation, respectively, across five runs.

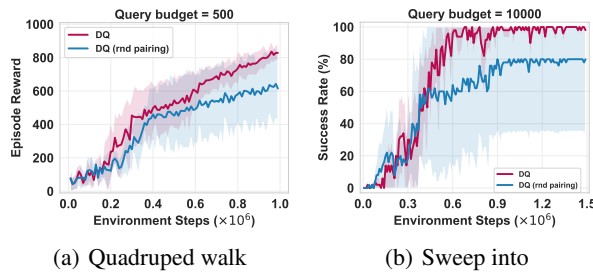

(a) Quadruped walk      (b) Sweep into

*Figure 6.* Ablation study evaluating different **PCQ pairing approaches**. Learning curves on locomotion tasks illustrate the comparison between our full method (DQ) and a variant employing random pairing (rnd pairing). The solid curves and shaded regions represent the mean and standard deviation, respectively, across five runs.

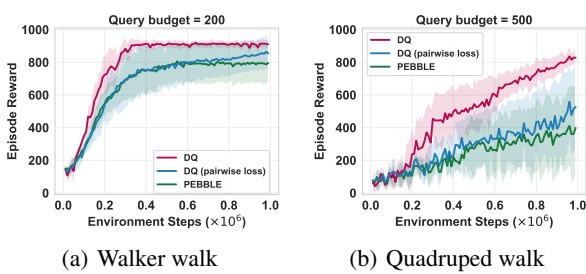

(a) Walker walk      (b) Quadruped walk

*Figure 7.* Ablation study evaluating different **loss functions**. Learning curves on locomotion tasks illustrate the comparison between our full method (DQ), a variant using only the pairwise loss, and the baseline PEBBLE. The solid curves and shaded regions represent the mean and standard deviation, respectively, across five runs.

## 5.5. Ablation Studies

To isolate the contributions of our proposed techniques to the overall performance of DistQ, we conduct ablation studies focusing on three key components: the query selection strategy, the PCQ pairing approach, and the newly designed loss function.

First, we examine the **query selection strategy**. Our full method (DQ, pink) prioritizes first selecting informative PCQs and then considers query easiness among those candidates. We compare this against three variants: a variant, EI (blue), which reverses this order by prioritizing easiness first and then informativeness; a variant, I (green), which selects PCQs based solely on informativeness; and a variant, E (orange), which selects PCQs based solely on easiness. Performance in terms of episode reward on two locomotion tasks is presented in Figure 5. To complement this, we also assess easiness of the PCQs selected by each variant by plotting the count of incorrectly predicted preference feedback in Figures 4(c) and 4(d), as discussed in Section 5.3.

The EI variant (blue), which prioritizes easy queries before informative ones, shows a significant decrease in episode reward on the considered tasks (Figure 5). This outcome suggests that an "easy first" approach can be detrimental as it may not provide sufficient information for robust reward learning. When selecting queries based only on informativeness (I, green), the agent achieves strong, sometimes near-optimal, reward performance. However, this variant consistently generates the highest number of queries that are incorrectly predicted (Figures 4(c) and 4(d)), indicating that these queries would likely be most difficult for the labeler. Conversely, selecting solely for easiness (E, orange) results in the fewest incorrect predictions (i.e., the easiest queries for the labeler) but leads to the poorest reward performance. These findings highlight a clear trade-off. DQ (pink) effectively balances these aspects. By first ensuring informativeness and then selecting for easiness from that informative pool, DistQ achieves strong performance while presenting queries that are demonstrably easier to answer than those from the informativeness-only approach and more effective for learning than the "easy first" EI variant.

Second, we investigate the **PCQ pairing approach** used to construct DQs. We compare our proposed pairing approach against a random pairing baseline ("rnd pairing"). As shown in Figure 6, our pairing method significantly outperforms random pairing. This result underscores that the method used to combine PCQs into DQs influences their effectiveness, with our approach yielding DQs that better leverage human feedback for learning.

Finally, we ablate the components of our **newly designed loss function**, $\mathcal{L}_{\text{DQ}}^{\text{Rew}}$ (from Equation (9)), which integrates distinguishability preference feedback ($\mathcal{L}_d^{\text{Rew}}$) with standard pairwise preference feedback ($\mathcal{L}_{\text{PCQ}}^{\text{Rew}}$). To assess the contribution of $\mathcal{L}_d^{\text{Rew}}$ to the final performance, we compare three configurations in Figure 7: DistQ using the full $\mathcal{L}_{\text{DQ}}^{\text{Rew}}$ (pink), DistQ trained using only $\mathcal{L}_{\text{PCQ}}^{\text{Rew}}$ (blue), and the baseline PEBBLE (green), which also uses only a pairwise preference loss. As anticipated, removing $\mathcal{L}_d^{\text{Rew}}$ (the blue variant) impairs the performance compared to the full DistQ (pink), confirming the utility of distinguishability feedback. Surprisingly, though, DistQ trained only with $\mathcal{L}_{\text{PCQ}}^{\text{Rew}}$ (blue) still outperforms PEBBLE (green). Given that both configurations rely solely on the same number of pairwise comparison queries, this indicates that the queries selected by DistQ are more effective for reward learning. We provide results of above ablation studies on more tasks in Appendix D.

## 6. Discussion

We introduce DistQ, a novel framework for RLHF designed to enhance both query efficiency and user-friendliness. At its core, DistQ features the *distinguishability query*, a new type of human interaction, coupled with an efficient and user-friendly query selection method that strategically balances the informativeness and easiness of queries. These key features, supported by a specifically designed loss function for reward learning, allow DistQ to learn more effectively from human feedback while minimizing the labeling effort. Experiments on a variety of locomotion and robotic manipulation tasks demonstrate that DistQ outperforms current state-of-the-art baselines in RLHF for control, particularly when considering the dual objectives of query efficiency and user-friendly query design.

**Limitations** While DistQ demonstrates strong performance, future work could further extend the method and its evaluation and apply it to new domains such as language modeling. The current measures for informativeness and easiness, though effective, could potentially be refined or replaced with more sophisticated metrics, which might further optimize query selection. Furthermore, our present evaluation is focused on simulated control tasks. Extending the validation of DistQ to more realistic and diverse real-world applications would provide further insights into its capabilities and robustness.

**Conclusion** In summary, DistQ offers a step towards more practical RLHF by directly addressing the need for queries that are both highly informative for the agent and easy to answer for human labelers. By advancing these dual objectives, we believe that DistQ contributes a valuable perspective and a robust set of tools to develop more efficient and user-friendly RLHF systems.

# Acknowledgements

This work has been supported in part by the National Key R&D Program of China (Grant No. 2024YFC3017100), the program of National Natural Science Foundation of China (No. 62176154) which is a research project funded by NetEase, and the LMUexcellent project funded by the Federal Ministry of Education and Research (BMBF) and the Free State of Bavaria under the Excellence Strategy of the Federal Government and the Länder as well as by the Hightech Agenda Bavaria. Eyke Hüllermeier has received funding from the European Union's Horizon Europe research and innovation programme under the Marie Sklodowska-Curie grant agreement No. 101073307. We also thank anonymous reviewers for critically reading the manuscript and suggesting substantial improvements.

# Impact Statement

This paper presents work whose goal is to advance the field of Machine Learning. There are many potential societal consequences of our work, none of which we feel must be specifically highlighted here.

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

# A. Baselines

All baseline methods considered in this paper are under the RLHF framework as we explained in Section 3. In this section, we provide more details about these baselines. Specifically, we summarize the query selection methods adopted by DistQ and baselines in Appendix A.

*Table 1.* Query selection strategies employed by different methods.

| Method | Query Selection Strategy |
|--------|--------------------------|
| DistQ | Larger variance + lower entropy |
| PEBBLE | Larger entropy |
| SURF | Larger variance |
| RUNE | Larger variance |
| MRN | Larger variance |
| QPA | Random selection from near-on-policy buffer |

## A.1. PEBBLE

PEBBLE (Lee et al., 2021b) refines the preference-based reward learning paradigm introduced by Christiano et al. (2017) by substituting the original on-policy RL algorithm with the off-policy algorithm SAC. It further enhances the sample efficiency and stability of the preference-based RL framework (as described in Section 3) through three primary mechanisms:

1. **Unsupervised Pre-training:** To generate diverse initial trajectories and accelerate subsequent reward learning, PEBBLE pre-trains the policy by maximizing the entropy of states encountered during exploration. This contrasts with random initialization and aims to provide a richer starting point for learning from preferences.

2. **Entropy-based Query Selection:** To select informative PCQs for human labeling, PEBBLE first samples a large batch of candidates. It then preferentially selects pairs that exhibit higher predictive entropy ($\mathcal{H}(P_\psi)$) from the current reward model, prioritizing queries where the reward model is most uncertain.

3. **Replay Buffer Relabeling:** To ensure consistent reward signals for the off-policy SAC algorithm, which can be sensitive to non-stationary rewards, PEBBLE periodically re-evaluates and updates the reward labels for transitions stored in its replay buffer using the latest learned reward model.

## A.2. SURF

SURF (Park et al., 2022), building upon the PEBBLE framework, aims to reduce the quantity of human feedback required in RLHF while maintaining or even enhancing task performance. This is achieved by introducing data augmentation techniques designed to maximize the utility of available data. Specifically, SURF employs a semi-supervised learning (SSL) approach to generate pseudo-labels for unlabeled trajectory segments and proposes a tailored data augmentation strategy for PCQs. These mechanisms collectively enable SURF to significantly improve the query-efficiency of RLHF.

## A.3. RUNE

RUNE (Liang et al., 2022), which also integrates with the PEBBLE framework, introduces an intrinsic reward mechanism to enhance exploration. This intrinsic reward quantifies novelty by measuring the disagreement across an ensemble of learned reward models. By incorporating the uncertainty of reward models directly into the agent's total reward signal, RUNE incentivizes the agent to explore regions of the state space where the reward function is least certain, thereby encouraging better exploration.

## A.4. MRN

MRN (Liu et al., 2022) is a data-efficient RLHF framework that uses bi-level optimization for concurrent reward and policy learning, learning the $Q$-function and the policy at the inner level while adapting the reward model to the $Q$-function performance at the outer level. Specifically, MRN operates in two nested loops:

- **Inner Loop:** The agent learns a $Q$-function and a corresponding policy using a conventional RL algorithm, based on the current estimate of the reward function.

- **Outer Loop:** The reward function is adaptively updated to improve the consistency of the $Q$-function with the collected human preference data.

By optimizing the reward function based on its downstream impact on policy performance with respect to human preferences, MRN aims to significantly improve data efficiency.

### A.5. QPA

QPA (Hu et al., 2024) addresses the issue of query-policy misalignment observed in PEBBLE. The authors find that selecting PCQs solely to maximize the overall quality of the learned reward model may not translate to improved policy performance, as such queries might not be relevant to the agent's current learning needs. QPA tackles this by enforcing a bidirectional alignment between the queries presented to the human and the agent's evolving policy. This is primarily achieved through two components:

- **Policy-Aligned Query Selection:** Queries are preferentially selected from recent trajectories generated by the current policy. This ensures that human feedback is directly relevant to the agent's ongoing behavior, making the feedback more impactful for policy learning.

- **Hybrid Experience Replay:** A specially designed experience replay mechanism is used, which prioritizes recent, on-policy experiences.

## B. Experimental Settings

| HYPERPARAMETER | VALUE | HYPERPARAMETER | VALUE |
|---|---|---|---|
| **General settings** | | | |
| Initial temperature | 0.1 | Hidden units per each layer | 1024(DMControl) |
| | | | 256(Meta-world) |
| Length of segment | 50 | # of layers | 2(DMControl) |
| | | | 3(Meta-world) |
| Learning rate | 0.0003 (Meta-world) | Batch Size | 1024(DMControl) |
| | 0.0005 (Walker) | | 512(Meta-world) |
| | 0.0001 (Quadruped, Humanoid) | Optimizer | Adam |
| Critic target update freq | 2 | Critic EMA $\tau$ | 0.005 |
| $(\beta_1, \beta_2)$ | (0.9,0.999) | Discount $\bar{\gamma}$ | 0.99 |
| Frequency of feedback | 5000 (Meta-world, Humanoid) | Maximum budget / | 10000/50, 500/50, 200/20 (DMControl) |
| | 20000 (Walker) | # of queries per session | 10000/50, 3000/30, |
| | 30000 (Quadruped) | | 2000/100, 400/10 (Meta-world) |
| # of ensemble models $N_{en}$ | 3 | # of pre-training steps | 10000 |
| **Other settings for DistQ** | | | |
| Loss weights $(\lambda_d, \lambda_p)$ | (1, 1) | Size of $\mathcal{Q}_p$ | 10×# of queries per session |
| Size of $\mathcal{Q}_{pv}$ $(n_I)$ | 5×# of queries per session | Size of $\mathcal{Q}_{pve}$ $(2n_E)$ | 2×# of queries per session |

*Table 2.* Hyperparameters setting

### B.1. Implementation Details

DistQ is implemented based on the PEBBLE framework. DistQ and all the baseline methods follow the general hyperparameter configurations described in Appendix B. For other specific hyperparameter settings of the baselines, we follow the corresponding publications and published source code.

For the evaluation tasks, we choose the ones that were adopted by the baselines with their corresponding settings of the querying process. This choice is motivated by two considerations: First, given the complex framework combining reward learning and RL training, it may take a lot of effort to tune the hyperparameters for each method to work for new tasks. Second, we try to ensure a fair comparison with baselines by avoiding the possibility of choosing specific tasks and tuning hyperparameters that are beneficial to our method. In addition, this showcases the robustness of our method to some extent.

### B.2. Computational Resources

For all experiments, we only need one GPU card to launch experiments. Experiments were carried out on different platforms, including GeForce RTX 3060 12G GPU + 48GB memory + Intel Core i7-10700F, or GeForce RTX 3060 12G GPU + 64GB memory + Intel Core i7-12700, or GeForce RTX 2070 SUPER + 32GB memory + Intel Core i7-9700, or GeForce RTX 2060 + 64GB memory + Intel Core i7-8700.

Note that the training time for one specific method varies when run on different platforms. However, given the same computational platform, all evaluated methods take similar training time (no more than 12 hours even for the task requiring most queries). That is, introducing DistQ does not bring about excessive additional computational cost.

## C. Algorithm

---

**Algorithm 1** DistQ

---

1: Randomly initialize policy model $\pi_\phi$ and reward model $\hat{r}_\psi$
2: Dataset for trajectories $\mathcal{B} \leftarrow \emptyset$
3: Dataset for distinguishability feedback $\mathcal{Q}_{\mathrm{DQ}} \leftarrow \emptyset$
4: //PRE-TRAIN
5: Pre-training as PEBBLE's to obtain $\mathcal{B}, \pi_\phi$
6: **for** each iteration **do**
7:   //REWARD LEARNING
8:   **if** Iteration$\%K == 0$ **then**
9:     //SAMPLING QUERIES
10:     Randomly sample segment pairs $\mathcal{Q}_p = \{(\sigma^0, \sigma^1)\}$ from $\mathcal{B}$
11:     Calculate informativeness $\mathcal{I}$ of PCQs in $\mathcal{Q}_p$ (Equation (5)). $\mathcal{Q}_{pv}$ consists of the top $n_I$ PCQs with larger $\mathcal{I}$
12:     Calculate easiness $\mathcal{E}$ of PCQs in $\mathcal{Q}_{pv}$ (Equation (6)). $\mathcal{Q}_{pve}$ consists of the top and bottom $n_E$ PCQs in terms of $\mathcal{E}$
13:     Sort PCQs in $\mathcal{Q}_{pve}$ according to $\mathcal{E}$, and pair the $i$-th with the $(i + n_E)$-th to form $\mathcal{Q}_d$
14:     $\mathcal{Q}_{\mathrm{DQ}}' \leftarrow \mathcal{Q}_d$ with feedback
15:     $\mathcal{Q}_{\mathrm{DQ}} \leftarrow \mathcal{Q}_{\mathrm{DQ}} \cup \mathcal{Q}_{\mathrm{DQ}}'$
16:     //TRAINING $\hat{r}_\psi$ ON EXTENDED $\mathcal{Q}_{\mathrm{DQ}}$
17:     **for** each gradient step **do**
18:       Randomly sample a minibatch $\{((\sigma_0^0, \sigma_0^1), (\sigma_1^0, \sigma_1^1), \boldsymbol{y}^{\mathrm{DQ}})\}$ from $\mathcal{Q}_{\mathrm{DQ}}$
19:       Optimize $\mathcal{L}_{\mathrm{DQ}}^{\mathrm{Rew}}$ ( Equation (9)) with respect to $\psi$
20:     **end for**
21:   **end if**
22:   //COLLECT TRAJECTORIES
23:   **for** each timestep $t$ **do**
24:     Collect interaction data by $a_t \sim \pi_\phi(a_t|s_t), s_{t+1} \sim P(s_{t+1}|s_t, a_t)$
25:     Store $\mathcal{B} \leftarrow \mathcal{B} \cup (s_t, a_t, s_{t+1})$
26:   **end for**
27:   //AGENT LEARNING
28:   **for** each gradient step **do**
29:     Optimize $\pi_\phi$ with a minibatch $\{(s, a, \hat{r}_\psi(s, a), s')\}$ randomly sampled from $\mathcal{B}$
30:   **end for**
31: **end for**

---

## C.1. Pseudo-Code

Algorithm 1 lists the pseudo-code of our algorithm, differentiating between PEBBLE and our changes with the use of color (changes in orange).

# D. Additional Experimental Results and Analysis

## D.1. Numerical Experimental Results

To compare the performance of DistQ and other baseline methods shown in Figure 3 more precisely, we list average values of the final performance over five runs for each method in Tables 3 and 4, along with the corresponding standard deviation shown in parentheses. For each task, the **highest average performance** is shown in **bold** and the *second highest* is in *italic*, except for the upper bound SAC.

Note that the table only shows the final performance of each method. The overall performance needs to be compared via both the learning curves and the numerical values. Tables 3 and 4 show that DistQ with full budget always achieves the best average final performance compared with other baselines on all considered tasks except Humanoid stand, which is consistent with Figure 3. In addition, DistQ with half budget also realizes superior or competitive performance considering all the other baselines on different tasks.

| Task | Walker walk [200] | Quadruped walk [500] | Humanoid stand [10000] |
|------|-------------------|----------------------|------------------------|
| SAC | 968.42 (8.09) | 894.08 (58.34) | 542.32 (55.72) |
| DistQ | **909.93 (25.22)** | **828.67 (56.95)** | *420.76 (60.15)* |
| DistQ (half) | *891.64 (72.68)* | 538.25 (236.65) | 251.44 (198.28) |
| PEBBLE | 795.28 (127.04) | 398.45 (288.37) | 233.66 (208.78) |
| SURF | 756.56 (110.78) | 257.82 (78.70) | 6.69 (1.66) |
| RUNE | 602.69 (301.13) | 306.57 (240.40) | 298.15 (159.73) |
| MRN | 877.63 (49.05) | *615.08 (207.24)* | 261.18 (233.49) |
| QPA | 874.97 (193.05) | 413.22 (298.14) | **456.17 (54.38)** |

*Table 3.* Episode reward on locomotion tasks, with query budget denoted in the square brackets, corresponding to Figures 3(a) to 3(c).

| Task | Window open [400] | Door unlock [2000] | Door open [3000] | Sweep into [10000] | Disassemble [10000] |
|------|-------------------|--------------------|--------------------|---------------------|---------------------|
| SAC | 1.00 (0.00) | 1.00 (0.00) | 1.00 (0.00) | 1.00 (0.00) | 0.00 (0.00) |
| DistQ | **0.98 (0.04)** | **0.98 (0.04)** | **1.00 (0.00)** | **0.98 (0.04)** | **0.20 (0.44)** |
| DistQ (half) | 0.78 (0.43) | 0.78 (0.43) | *0.94 (0.13)* | 0.78 (0.43) | *0.18 (0.40)* |
| PEBBLE | 0.54 (0.30) | 0.50 (0.46) | 0.46 (0.50) | 0.70 (0.40) | 0.00 (0.00) |
| SURF | *0.80 (0.44)* | *0.92 (0.17)* | 0.60 (0.54) | 0.58 (0.53) | 0.02 (0.04) |
| RUNE | *0.80 (0.39)* | 0.86 (0.19) | **1.00 (0.00)** | *0.96 (0.08)* | 0.00 (0.00) |
| MRN | 0.28 (0.42) | 0.50 (0.47) | 0.80 (0.44) | 0.78 (0.43) | 0.00 (0.00) |
| QPA | *0.80 (0.18)* | 0.69 (0.33) | 0.52 (0.45) | 0.55 (0.41) | 0.008 (0.008) |

*Table 4.* Success rate on manipulation tasks, with query budget denoted in the square brackets, corresponding to Figures 3(d) to 3(h).

## D.2. Results of Ablation Study

In this section, we provide more results of the ablation study in Figure 8 as mentioned in Section 5.5.

## D.3. Detailed User Study

This section provides implementation details for the user study conducted in the Quadruped environment. Since DistQ builds on PEBBLE, an online RLHF framework, we implemented our own user study pipeline rather than using existing platforms designed for offline RLHF (Yuan et al., 2024). While developing systematic platforms for online RLHF remains

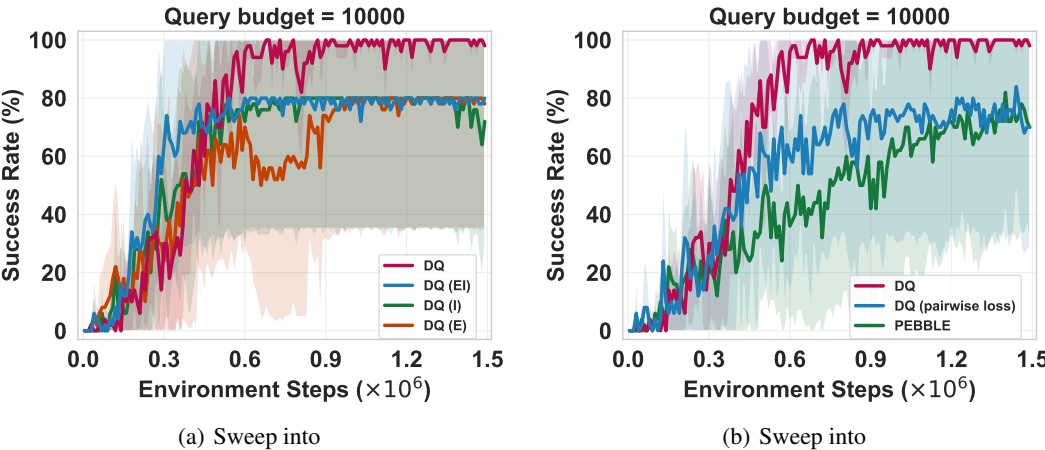

(a) Sweep into                      (b) Sweep into

*Figure 8.* Learning curves of more ablation studies on the manipulation task. The solid curves and shaded regions represent the mean and standard deviation, respectively, across five runs.

an open problem for future work, our custom implementation enables direct evaluation of our method in human feedback scenarios.

In our designed user study, the human labeler aims to train the agent to stand and wave its right hind leg. For both DistQ and PEBBLE, the labeler is asked 150 queries in total by the agent, which take about 3 hours to answer during the online interleaving of reward learning and agent learning. We evaluate the learned behavior of the agent at the end of training for both methods. Results show that with DistQ, the agent can successfully perform the desired behavior. However, with PEBBLE, it hardly even stands up. Videos of selected queries and evaluation of trained agents for both methods are available at https://zenodo.org/records/15606992.

