# OpenReview forum: "Comparing Comparisons: Informative and Easy Human Feedback with Distinguishability Queries"
_ICML.cc/2025/Conference — ICML 2025 poster_

### Official Review · Reviewer_6jG4 · 2025-03-12

**Overall Recommendation:** 3

**Summary:**

This paper proposes a new type of query in rlhf, the distinguishability query (DQ). Rather than directly comparing two sets of trajectories, the authors compare two sets of trajectories and selects the one that is easier to give feedback on. They then provide feedback on the easier pair, and can learn from that data. Experimental results show that the method can sometimes produce solid performance gains over the baseline, PEBBLE, but when equal amounts of data are used for the baseline and DQ the performance gain is not that large.

## After Discussion
After the discussion and reading the reviews of the other reviewers, I plan to maintain my score. The method is interesting and the paper seems solid overall. However, it is still unclear to me how to measure the cost of a distinguishability query compared to a normal RLHF query and what a fair comparison means for DQ versus other baselines. I don't really think the authors argument that "it's also unfair to only count the number of human choices since such choices for DQ and PCQ apparently provide different amount of information" makes sense, which is the essential claim all of their experimental results rely upon.

**Claims And Evidence:**

The main claim that DQ can improve RLHF performance is somewhat supported. The experimental results show that DQ can outperform the baseline algorithms when it recieves a large budget then them (see DQ in figure 3). But when a roughly equal labeling budget is given for the baselines and DQ, (i.e. DQ (Half)), the results are not that good and can only really outperform the baseline in two of the four environments presented in Figure 3.

In the experiments section the authors claim that DQ (Half) outperforms the baselines in all except for MRN on Quadruped walk (line 345). However, this is a but overstated. DQ-half and other algorithms typically have very similar performance (at least in Figure 3) as well as overlapping confidence intervals.

**Essential References Not Discussed:**

I do not know of any.

**Experimental Designs Or Analyses:**

Yes I did check the soundness of the experimental design. The experimental design and analysis seems both standard and solid. They compare their algorithm with many baselines, and they are careful to make sure the data labelling budget is comparable for different algorithms. In addition, they report five independent runs for each algorithm, and report standard deviation for all experiments.

**Methods And Evaluation Criteria:**

The experiments and evaluation criteria do make sense. This paper works to improve RLHF, and some of the earliest papers on RLHF focused on continuous control [1]. They evaluate based on the average ground-truth reward received, which makes sense. However their experiments are all conducted on very simple continuous control environments. My concern is that their approach of only selecting easy samples to train on will work better in easy settings than in hard settings. This means their experimental setup may overestimate the utility of their method.

[1] Christiano, Paul F., et al. "Deep reinforcement learning from human preferences." Advances in neural information processing systems 30 (2017).

**Other Comments Or Suggestions:**

none

**Other Strengths And Weaknesses:**

Strengths:
- The idea of this paper is novel and interesting.
- The writing and motivation for their method is very clear.
- The experimental results are for the most part solid.

Weaknesses:
- The problem settings used in this paper are very easy problems. In continuous control we have a fairly straightforward goal, so it is fine to rely on non-ambiguous queries to learn it. However for more complex and open ended tasks such as LLM alignment, ambiguous queries may actually contain subtle and important information for reward learning.
- DQ (half) does not really perform much better than most baselines.

**Questions For Authors:**

Is there any way to test DistQ on LLM Alignment? Or, is it possible to consider a task where subtle information in ambiguous queries matters?

**Relation To Broader Scientific Literature:**

The idea of distinguishing between different queries is definitely a novel idea in RLHF. The paper does a good job of engaging with the literature.

**Theoretical Claims:**

na

---

> ### Author Rebuttal · Authors · 2025-04-01
>
> 1. Claim about performance of DQ (half):
>
> (1) For query budget and fairness comparison discussion, please refer to **point 1-(1)&(2) in our response to reviewer ufMA**.
>
> (2) For effectiveness, please refer to **point 1-(3) in our response to reviewer ufMA**. Besides, we show in Fig 2 in https://drive.google.com/drive/folders/1wR469npWztzkjyW0YF2H9C10LTI3wnTU that DistQ learning from only distinguishability preference feedback (DQ_d loss) performs far behind DistQ learning from only pairwise preference feedback (DQ_pairwise loss), which demonstrates that DQ (half) actually obtains much less information from a seemingly "roughly equal labeling budget" compared with the baselines. In such case, performance of DQ (half) shown in Fig 3 could be recognized.
>
> At last, to clarify, the comment in line 345 saying DQ (half) "outperforms most baselines only except for MRN" is only made on Quadruped walk", which we think is consistent with Fig 3. As for other tasks, DQ (half) indeed has similar performance to some of the baselines, which is understandable given the above explanation.
>
> 2. Overestimation from easy experimental setup:
>
> (1) We've conducted more experiments on harder control tasks. Please refer to **point 1 in our response to reviewer pH8H and mQWc**. Besides, as pointed out by reviewer mQWc, there also exist challenging control tasks to solve. We believe that for those tasks, subtle information in ambiguous queries does matter, and we demonstrate that our method can work.
>
> (2) Note that our method balances query efficiency and user-friendliness by selecting both informative (ambiguous) and easy-to-answer queries as introduced in Sec 4.2, instead of only selecting easy samples to train on. In our experiments (see Sec 5.5), we also conduct an ablation study to compare the performance of our method and an alternative approach that samples queries only based on easiness (DQ (E) curve in plots). The results show that our method considering both ambiguity and user-friendliness is much better than DQ (E).
>
> 3. About testing DistQ on LLM Alignment:
>
> In our current setting for DistQ, the reward model is an ensemble of three-layer neural networks with 256 hidden units, which is quite small compared with the reward model in LLM alignment. Therefore, it may need much more queries and computational resources for LLM alignment, which is hard for us to test given the limited time.
> We believe the high-level idea of DistQ can be applied for LLM alignment, and we leave this extension to future work.

---

> > ### Comment · Reviewer_6jG4 · 2025-04-08
> >
> > Thank you for the detailed response to my review. I appreciate the addition of harder experiments. For figure 2 of the rebuttal material, how do you decide on the budget? Does a difficulty query use the same amount of budget as a normal query?

---

> > > ### Author Response · Authors · 2025-04-09
> > >
> > > Thank you for taking the time to engage in this discussion with us. We are also very grateful for your appreciation of our efforts on additional experiments. We now provide results on more tasks (**please refer to point 2 in our reply to reviewer pH8H's comments**), which further demonstrates the effectiveness of our method.
> > >
> > > For figure 2 of the rebuttal material, we decide the budget based on our estimation of the difficulty level of tasks. For example, given the hard task Disassemble, we guess that its difficulty level is similar to or larger than Sweep into (which is the hardest task in our original paper) based on our understanding of the tasks. Then we set the query budget for Disassemble as 10,000, which is the query budget we use for Sweep into. Though from the results, it seems that such budget may be insufficient for Disassemble. However, given the limited time, we are not allowed to try a larger budget.
> > >
> > > We sincerely hope that our response can address your concerns and demonstrate our method better. If no other concern, we would be grateful if you could consider increasing your evaluation of our work.

---

### Official Review · Reviewer_mQWc · 2025-03-14

**Overall Recommendation:** 3

**Summary:**

This paper proposes a three-stage approach to optimizing the PbRL pipeline:

1. **Selecting the top N informative (based on the variance of reward ensembles) Pairwise Comparison Queries (PCQs)**  for comparison. This step explicitly enables the reward model to distinguish high-uncertainty pairs better, accelerating the reward learning process.

2. **Selecting the top and bottom M easy (based on entropy) PCQs** for re-pairing. This enhances the contrast of comparison pairs, thereby improving the sample efficiency of reward learning.

3. **Incorporating an additional distinguishability preference loss** to assist reward learning further.

Intuitively, all these improvements contribute positively to the PbRL pipeline. Comparative and ablation experiments demonstrate the effectiveness of these stages.

## update after rebuttal
During the rebuttal phase, the author supplemented many materials through anonymous links, such as new experiments on harder tasks, videos of learned policies, details of the user study, and explained some misunderstandings in the discussion section, which led me to increase my score.

**Claims And Evidence:**

Although the methods proposed in this paper are intuitively reasonable, I believe the experiments are insufficient to support these claims.

- The experiments are conducted on a very limited set of tasks, including two locomotion tasks and two manipulation tasks. Among them, Walker Walk and Quadruped Walk are relatively easy tasks in DMC, while Window Open and Sweep Into correspond to easy and medium-difficulty tasks in MetaWorld, respectively. If the authors could include more challenging tasks, such as **Humanoid/Dog** in DMC and **Shelf Place/Disassemble** in Meta-World, the experiments would be more convincing.
- As a PbRL algorithm, the experiments do not involve **human feedback**. Evaluating only on synthetic feedback fails to demonstrate the algorithm's effectiveness in real-world scenarios.

**Essential References Not Discussed:**

No.

**Ethical Review Concerns:**

No.

**Experimental Designs Or Analyses:**

See **Claims And Evidence**.

**Methods And Evaluation Criteria:**

Yes. However, I suggest that the authors include additional experiments that go beyond merely maximizing reward and instead use human feedback to shape the agent's behavior, as Pebble did in Figure 6 of its paper.

**Other Comments Or Suggestions:**

No.

**Other Strengths And Weaknesses:**

See **Claims And Evidence**.

**Questions For Authors:**

I have no questions.

**Relation To Broader Scientific Literature:**

The key contributions of the paper are mainly related to preference-based RL.

**Theoretical Claims:**

This paper includes no proofs or theoretical claims.

---

> ### Author Rebuttal · Authors · 2025-04-01
>
> 1. About experiments on more challenging tasks:
>
> **Please refer to point 1 in our response to reviewer pH8H**. Besides, it is worth mentioning that the suggested harder tasks are not evaluated in all our baselines either. Therefore, we need to determine workable hyper-parameters for all the methods as well as the backbone RL algorithm SAC, which can be time-consuming given the high difficulty of tasks. In our linked results, all methods follow their default hyper-parameter settings without tuning.
>
> Also, there is possibility that these tasks are already challenging when a ground-truth reward function is available. Our method doesn't claim to improve the sample efficiency of the underlying deep RL algorithm. In contrast, we propose a method to circumvent the need of having to define a ground-truth reward function, by asking informative and easy-to-answer queries than other RLHF methods.
>
> 2. Additional experiments with real human feedback:
>
> We have conducted the suggested experiments with real human involved and explained details in Sec 5.4 and Appendix D.3. To have a comprehensive understanding of the user study, we also provided an anonymous link of videos of selected queries and evaluation of trained agents in Appendix D.3 in our original version of paper.

---

> > ### Comment · Reviewer_mQWc · 2025-04-07
> >
> > First, I would like to thank the authors for the additional experiments on more challenging tasks and the user study.
> >
> > I am now also considering Reviewer ufMA’s perspective. From a practical standpoint, **understanding the video pair is the most time-consuming part** of the human feedback process. Once a video pair is understood, it is not difficult to make even multiple choices. From this perspective, DistQ(half) requires the annotator **to fully understand the easy pair and only briefly understand the hard one**, whereas DistQ requires the annotator **to fully understand two video pairs**. This makes the comparison between DistQ(half) and the baselines relatively fair, while DistQ consumes twice the query budget compared to the baselines.  Unfortunately, DistQ(half) does not show a clear advantage over the baselines.
> >
> > However, from the additional experiments, I notice that DistQ(half) performs significantly better than the baselines in the *disassemble*. This makes me wonder whether the tasks in the current experiments are too simple, making it difficult for DistQ(half) and the baselines to show a noticeable difference.

---

> > > ### Author Response · Authors · 2025-04-08
> > >
> > > Thank you for acknowledging our response. Meanwhile, we would like to emphasize that **the user study was done in our paper originally, instead of during the rebuttal**. We next address the further concerns and clarify potential misunderstanding.
> > >
> > > 1. About time consumption of human feedback
> > >
> > > We agree that understanding the video pairs is time-consuming, especially for harder ones. And **this exactly matches our core motivation**. Our method is proposed to select informative and relatively easier-to-answer PCQs. **For one DQ in both DistQ and DistQ(half) settings** (see our detailed explanation in the following **point 2**), **the annotator only needs to select and answer the easier PCQ, thus avoiding fully understanding the more complex PCQ**.
> > >
> > > Besides, we believe that reviewer ufMA didn't discuss about understanding the video pair and its time consumption for the human feedback process. Instead, he/she presented his/her understanding of *number of human choices for each query*.
> > > The suggested references of human response time serve as a supplement to our relevant work section.
> > >
> > > 2. About the experimental settings of DistQ and DistQ(half)
> > >
> > > **Please refer to point 1-(1) in our response to reviewer ufMA.** We believe that our explanation clarifies potential misunderstanding. Specifically, for both DistQ and DistQ(half), the human chooses the preferred trajectory only from the chosen PCQ in a DQ. For each DQ in both settings, the human always makes 2 choices instead of 3.
> > > In this case, the annotator doesn't have to fully understand the two video pairs, e.g., s/he only has to understand the easier one, and doesn't need to spend a lot of effort to understand the harder one. Therefore, **DistQ shares the same requirement as DistQ(half) for one DQ, instead of requiring "the annotator to fully understand two video pairs"**.
> > >
> > > 3. About the fairness in our experimental evaluation
> > >
> > > Given our above clarification, DistQ actually doesn't consume twice the query budget.
> > > For one DQ, the human makes 2 choices, which may be considered unfair when comparing with baselines using PCQs where the human only makes 1 choice. That's why we design DistQ(half), which only uses a half number of DQs (=equal number of human choices as baselines). However, it's also unfair to only count the number of human choices, since such choices for DQ and PCQ apparently provide different amount of information. Therefore, we consider DistQ with full budget.
> > > **Please also refer to point 1-(2) in our response to reviewer ufMA, and also our explanations in the first paragraph of Sec 5.2 in our paper**.
> > >
> > > 4. About the effectiveness of DistQ(half)
> > >
> > > **Please refer to point 1-(3) in our response to reviewer ufMA**. Based on our above explanation, though DistQ(half) receives the same number of human choices (for both DQs and PCQs) as other baselines (for only PCQs), the information DistQ(half) obtains is much less than the baselines. **Please also refer to our discussion in point 1 to reviewer 6jG4**. Even though, DistQ(half) still achieves at least a similar performance, while only requiring the annotator to fully answer relatively easier PCQs, which can highly demonstrate its effectiveness.
> > >
> > > 5. About the performance of DistQ(half) in disassemble
> > >
> > > Please refer to **point 1 in our first-round response** to you. The suggested harder tasks were not evaluated in all baselines before. So we can't decide how many queries are needed for the baselines to work. It's possible that our tested query budget is enough for DistQ(half) to work a bit, but is still insufficient for the baselines to work.
> > >
> > > As for the simpler tasks in the paper, they are widely tested by the baselines and workable query budget is also provided. Given the smaller difficulty level of the tasks and enough query budget, it's possible that the performance difference between DistQ(half) and other baselines is smaller.
> > >
> > > ---
> > > We sincerely hope that all our responses have addressed your concerns and mitigated potential misunderstanding. If you have any other concerns, we would be happy to discuss them. If not, we would be grateful if you could consider increasing your evaluation of our work.

---

### Official Review · Reviewer_pH8H · 2025-03-16

**Overall Recommendation:** 4

**Summary:**

This paper proposes Distinguishability Queries (DistQ), a new method for improving RLHF. DistQ reduces cognitive load by letting humans first choose which of two trajectory comparisons is easier to evaluate and then provide feedback on the easier pair. This approach captures both preference strength and ordinal information, improving the efficiency of learning reward functions. Experiments show that DistQ is more data-efficient and user-friendly than traditional methods, offering a better approach for RLHF in tasks with complex objectives.

**Claims And Evidence:**

Yes, the claims presented in the text are all supported by experiments. I believe the aspect with some shortcomings is related to alleviating the burden on annotators, which may require more extensive user study.

**Essential References Not Discussed:**

n/a

**Experimental Designs Or Analyses:**

The experimental design is scientific and mainstream, aligned with the state-of-the-art PBRL community, and compares with strong baselines.

**Methods And Evaluation Criteria:**

Yes, the research question is very meaningful and compares a large number of benchmark algorithms, but the environments used are relatively limited.

**Other Comments Or Suggestions:**

see above

---
**After rebuttal comment:
Thank you for the author's response. I believe the additional experimental results and demonstrations can enhance the quality of the paper, so I vote to accept it and raise the score from 3 to 4. I hope the author can include these additions in the next revised version, adding more experimental content and demonstrations to the experimental section of the main text. This will help better present DistQ.**

**Other Strengths And Weaknesses:**

Strengths:
- The paper is written very elegantly, and the research problem is highly significant. Providing annotators with easy and better annotation methods is extremely important.
- The aggregation of the distq component is innovative. The experimental results show some improvement over previous methods, and the authors analyze the effects of some design choices.

Weaknesses:
- I believe in the outstanding performance of distQ, and it indeed compared with many baseline algorithms. However, why wasn’t it tested in more environments (in terms of quantity or type, 9 envs in b-pref totally)? I think this is more important than comparing with more baseline algorithms. If distq could be validated on a broader range of experimental benchmarks, its impact would be greater.
- For me, the most interesting aspect of this article is its ability to provide annotators with Informative and Easy Human Feedback. This is very important for RLHF in real-world scenarios, but the article does not delve deeper into this point through more analysis and experiments. Most experiments use synthetic feedback, which makes it difficult to validate this interesting perspective. Appendix D.3 provides simple experimental results, but I believe this deserves more discussion in the main text. This is a valuable approach.
- Which part or all parts of DistQ can be applied to broader domains? For example, real-world robotic arm experiments, offline RLHF experiments, or Atari/Minecraft.
- Lack of discussion on papers related to improving the quality of annotation human feedback and reducing the burden of human feedback, such as: [1][2][3][4]
[1] Yuan Y, et al. Uni-rlhf: Universal platform and benchmark suite for reinforcement learning with diverse human feedback[J]. arXiv preprint arXiv:2402.02423, 2024.
[2] Zhang L, et al. Crew: Facilitating human-ai teaming research[J]. arXiv preprint arXiv:2408.00170, 2024.
[3] Dong Z, et al. Aligndiff: Aligning diverse human preferences via behavior-customisable diffusion model[J]. arXiv preprint arXiv:2310.02054, 2023.
[4] Metz Y, et al. Reward Learning from Multiple Feedback Types[J]. arXiv preprint arXiv:2502.21038, 2025.

**Questions For Authors:**

see above

**Relation To Broader Scientific Literature:**

n/a

**Theoretical Claims:**

n/a

---

> ### Author Rebuttal · Authors · 2025-04-01
>
> 1. About more experimental environments:
>
> For the current version of our paper, we selected representative control tasks of different difficulty levels (in terms of query budget needed to accomplish the task) and of different types (locomotion and robotic manipulation) to demonstrate the performance of our method. Following the reviewers' suggestions, **we now validate our method on more tasks, including the hard locomotion task Dog walk (hard), and robotic manipulation tasks Door open (easy) and Disassemble (hard)**. Results are shown in Fig 1 in this link (https://drive.google.com/drive/folders/1wR469npWztzkjyW0YF2H9C10LTI3wnTU). For the latter two tasks, our method can still perform better or competitively compared with other baselines. For the hard Dog walk task, however, all methods (including SAC with ground-truth reward function) fail to achieve it. A possible explanation is that, for such complex tasks, it takes a large number of queries and needs efforts to find workable hyperparameters for the backbone framework. Given the time limit, we haven't found proper settings for all methods and we've tried our best to test on as many tasks as possible. We will include more results in our final version if time permits.
>
> 2. About more analysis and experiments:
>
> We deeply appreciate your approval of our method. Besides synthetic feedback, we also conduct user study with real human providing feedback as discussed in Sec. 5.4. We provide both statistical results and cognitive feedback from the participants here, which evidently support our argument of effectiveness and user-friendliness of DistQ. Larger scale user study may take a large mount of human labor. We will find efficient ways to enrich our user study in future.
>
> 3. About application to broader domains:
>
> DistQ is built on top of video-based queries and online interleaving of reward learning and agent learning, so the whole method can be straightforwardly adapted to domains like robotic arm experiments and games. As for offline RLHF tasks, we may need to adjust the implementation of the query selection criteria. But the proposed query type and high-level ideas for query selection can still be used in such settings.
>
> 4. About discussion of more related papers:
>
> Thank you for suggesting these recent related works. The first two mainly propose efficient and expandable platforms for RLHF experiments with real humans, which can be adopted for efficient large-scale user study. Considering your concerns about more extensive user study, we may utilize such platforms to perform user studies to further validate DistQ. The remaining two instead focus on refining RLHF approaches from different points of view. We will include a discussion about them in our final version.

---

> > ### Comment · Reviewer_pH8H · 2025-04-07
> >
> > Thank you for the detailed response. I believe the additional experiments have improved the paper, however, all methods generally perform mediocre in particularly challenging environments, making it difficult to conduct a thorough comparison. I am glad the author was able to include more difficult experiments. I suggest completing the 6-9 official BPref tasks based on the QPA method, as these tasks are of moderate difficulty. Additionally, I think the major contribution of this paper is providing a new way to collect feedback, so better visualization and making it more convenient for other researchers to use are also necessary. How does this feedback method work, and why is it better? It can be made clearer through some small demos or system demonstrations.

---

> > > ### Author Response · Authors · 2025-04-09
> > >
> > > Thank you for taking the time to engage in this discussion with us. We are also very grateful for your applause of our contribution of providing a new way to collect feedback for RLHF approaches.
> > > We next address your further concerns.
> > >
> > > 1. About the performance of challenging tasks
> > >
> > > Thank you for your recognition of our effort on more challenging tasks.
> > > We agree that a thorough comparison on those harder tasks suggested by reviewer mQWc is very difficult. This is because such harder tasks are barely evaluated before in most of the other baselines.
> > > Therefore, **we need to determine workable hyper-parameters for all the methods, which can be unpredictable and time-consuming given the high difficulty of the tasks**. In our linked results, all methods follow their default hyper-parameter settings without tuning for an intuitive and fair comparison.
> > > In this case, **our method still obviously outperforms other baselines on the hard task Disassemble, demonstrating its potential of solving hard tasks to some extent**.
> > >
> > > 2. About evaluation on more tasks based on QPA
> > >
> > > Thank you for your proposition of experiments on more tasks based on the QPA method.
> > > The experimental results are shown in ICML25_rebuttal_figures_round2.pdf in the link (https://drive.google.com/drive/folders/1wR469npWztzkjyW0YF2H9C10LTI3wnTU). Following the hyperparameter settings of QPA, we see that our method can always outperform other baselines on Door open. On Door unlock, DistQ still outperforms other baselines with slightly better performance than SURF. DistQ(half) also achieves competitive performance compared with other baselines except for falling behind SURF, which is understandable given our explanation about the fairness of our experiment setting and and effectiveness of our method (**please refer to point 1-(1)(2)(3) in our response to reviewer ufMA and also point 1 to reviewer 6jG4**).
> > > For the harder task Humanoid stand, although worse than QPA (which is understandable since we adopt the settings of QPA), DistQ significantly outperforms the other baselines and DistQ(half) realizes acceptable performance. From all our experiments on harder tasks, we argue that proper hyperparameter settings are critical for RLHF methods to work, which may explain the superiority of QPA on Humanoid stand. Therefore, we believe that our method could perform better on those harder tasks if suitable hyperparameters are adopted.
> > >
> > > Given the limited time during discussion, we only make it for these 3 tasks. We'll include the results of all suggested tasks in our final version.
> > >
> > > 3. About visualization and application of our method
> > >
> > > We provided a link (https://drive.google.com/drive/folders/1qvf7hJ-a66bGeu1g0f9ALWRtP9UgzDk1?usp=sharing) to videos of selected queries along with human labels, and evaluation of trained agents of our method and one baseline method in **Appendix D.3 of our paper**, which serves as a clear visualization of the whole process.
> > > Besides, **we provided detailed explanation of our method (see Fig 1 & 2 and Sec 4 ) and demonstration of its advantanges (see Sec 5) in the main paper. We also provided necessary details such as pseudo code and hyperparameters of our method in the appendix for convenient reproduction**. We will open source our code after publication to ensure the reproducibility of all our experiments.
> > >
> > > We sincerely hope that all our responses have addressed all the points you raised. If you have any other concerns, we would be happy to discuss them. If not, we would be grateful if you could consider increasing your evaluation of our work.

---

### Official Review · Reviewer_ufMA · 2025-03-18

**Overall Recommendation:** 3

**Summary:**

* This paper proposes a novel human feedback type for RLHF and an algorithm allowing robots to learn reward functions from such human feedback.
  * The novel feedback is that the robot first gives a human 2 pairs of trajectories, has the human choose the pair that is easier to choose, and then has the human choose one trajectory from that pair. In this way, from the 1st choice, the robot can infer the relative preference strength between the 2 pairs. From the 2nd choice, the robot can infer the preference. The benefit is that the robot can understand the preference strength in addition to the preference.
  * The proposed algorithm, DistQ, allows a robot to learn from such feedback:
    * Reward learning
      * Given a trajectory buffer, the robot randomly chooses pairs of trajectories.
      * The robot chooses the top n1 informative pairs based on variance.
      * The robot chooses the top and bottom nE easy pairs based on entropy.
      * The robot forms all the pairs, each of which has an easy trajectory with a hard trajectory.
      * The robot sends all these pairs to the human for feedback and then uses the feedback to infer the reward function.
    * Agent learning
      * The robot uses the estimated reward function to do RL to optimize policy and collect trajectories into the buffer.
      * Then, go back to the first step to update the reward function
* The 1st experiment compared the RL performance, given a fixed budget, of the proposed method with 5 baseline methods in simulated robot locomotion and manipulation tasks. The proposed method with full budget (DistQ) outperformed all baselines, while the proposed method with half budget (DistQ(half)) outperformed some of the baselines. A user study is also conducted to show that DistQ is better than PEBBLE.
* The 2nd experiment compared the query easiness, given a fixed budget, of the proposed method with 5 baseline methods in simulated robot locomotion and manipulation tasks. Both the proposed method with full budget (DistQ) and the proposed method with half budget (DistQ(half)) outperformed all baselines.
* The 3rd ablation experiment shows that the algorithm design, including choosing n1 informative pairs, choosing nE easy and hard pairs, and assign an easy and a hard trajectories in one pair, are significant to the performance.

## update after rebuttal
I appreciate the authors' Rebuttal in addressing my concerns. I have adjusted my score accordingly.

**Claims And Evidence:**

* The key claim is that the proposed query and algorithm can improve RLHF for robotic tasks. The empirical result supports this.
* One limitation of the empirical result is about DistQ and DistQ(half). The proposed novel query requires the human to make 2 choices, first choosing the easy pair and then chooses the preferred trajectory in the pair. By contrast, the conventional query requires the human to make 1 choice, choosing the preferred trajectory in the pair. As a result, it is a bit tricky to compare the proposed algorithm with the proposed query with standard RLHF methods.
  * The paper compared the following 2 variations of the proposed algorithm against the baseline methods:
    * **DistQ(half)**: In each query, the human first chooses the easy pair, and then chooses the preferred trajectory in the chosen pair. And this query with 2 choices counts as 1 query in the "query budget" in the empirical study.
      * +: This is consistent with the definition of the proposed query (Sec.4.1).
      * -: Unfortunately, easy pairs contain a limited amount of information, as mentioned under Eq.6. Since the human only provides the preference feedback for the easy pair selected by the human, the overall performance could be limited. This might be why DistQ(half) does not seem to outperform many baseline methods in Fig.3.
    * **DistQ**: In each query, the human first chooses the easy pair, and then chooses the preferred trajectory in the chosen pair **and also the not-chosen pair**. This query with 3 choices counts as 1 query in the "query budget" in the empirical study.
      * +: This outperformed all baseline methods as in Fig.3.
      * -: However, this is not consistent with the definition of the proposed query (Sec.4.1). As a result, I think it is not fair to compare DistQ with the baseline methods.
  * In addition to the 2 variations considered by the authors in the empirical study, I think there is one more variation that could be interesting to consider:
    * **DistQ(half-half)**: In each query, the human first chooses the easy pair, and then chooses the preferred trajectory in the chosen pair. This query with 2 choices counts as **2 queries** in the "query budget" in the empirical study.
      * The reasoning behind this variation is that the human makes 2 choices, so consuming 2 units of the query budget. This reasoning is consistent with the paper's interest in the query easiness.
  * Summary
    * I think it is fair to compare **DistQ(half)** against baseline methods, but it does not seem to perform that well empirically.
    * I think it is a bit unfair to compare **DistQ** against baseline methods.
    * I think that the authors could consider **DistQ(half-half)**, which, in my opinion, is also fair to be compared against the baseline method. But I conjecture that it will not perform as well as **DistQ(half)**.

**Essential References Not Discussed:**

* For Sec.2's Eliciting Preference Strength, two recent works explores the time elicitation in bandits (in the form of human response times), which can be relevant:
  * Shvartsman, M., Letham, B., Bakshy, E., & Keeley, S. L. (2024, July). Response time improves gaussian process models for perception and preferences. In The 40th Conference on Uncertainty in Artificial Intelligence.
  * Li, S., Zhang, Y., Ren, Z., Liang, C., Li, N., & Shah, J. A. (2024). Enhancing Preference-based Linear Bandits via Human Response Time. Advances in Neural Information Processing Systems, 37, 16852-16893.

**Experimental Designs Or Analyses:**

The experiment design, user study, and analyses make sense.

**Methods And Evaluation Criteria:**

Method and evaluation criteria make sense. There are 2 limitations.
* The 1st limitation is already discussed in `# Claims And Evidence*`.
* The 2nd limitation is that Sec.5.3 measures the query easiness as to whether the robot's predicted feedback is consistent with the ground truth. This definition seems to define the easiness from the robot's view. However, the query easiness is supposed to be defined from the human's view. I think a better way to measure query easiness is to use ground truth preference entropy for this query as defined in Eq.6.

**Other Comments Or Suggestions:**

NA

**Other Strengths And Weaknesses:**

## Strength
* The problem is well-motivated
* The paper is well-written.
* The proposed query is creative.
* The paper also contains a real user study to validate the method.

**Questions For Authors:**

* In the user study (Sec.5.4), there are 10 rounds. Based on App.D.3, in each round, the human answers 150 queries. So, in total, there are 1500 queries? Also, I am curious why PEBBLE has a successful rate of 0, which seems strange. Could you share some insights on why this is the case?

**Relation To Broader Scientific Literature:**

The paper focuses on the robot RLHF, which is a hot topic and also a very relevant problem to machine learning, LLM, and robotics.

**Theoretical Claims:**

NA

---

> ### Author Rebuttal · Authors · 2025-04-01
>
> 1. About experimental settings:
>
> (1) To address potential misunderstanding in the review, we first clarify some terminologies we used in our paper. One distinguishability query (DQ) consists of 2 pairwise comparison queries (PCQs). The human first chooses the more distinguishable PCQ and then chooses the preferred trajectory in this PCQ (line 202). The "query budget" in our paper refers to the number of PCQs (line 359) since all our baselines use PCQs. In the experiments section, for both DistQ and DistQ (half), the human **chooses the preferred trajectory only from the chosen PCQ in a DQ**, instead of "also the not-chosen pair" as stated in the review. That is, for each query in both settings, the human always makes 2 choices instead of 3. The difference between DistQ and DistQ (half) is that DistQ uses the same "query budget" as other baselines while DistQ (half) only uses half of the budget (line 364). See the first paragraph of Sec 5.2 for detailed instructions. We think both settings are consistent with our proposition.
>
> (2) We then discuss the fairness in our experimental evaluation. As stated in the review, for one DQ, the human makes 2 choices. We understand that this may be considered unfair when comparing with baselines using PCQs where the human only makes 1 choice. That's why we design DistQ (half), which only uses 100 DQs (=200 human choices). However, it's also unfair to only count the number of human choices since such choices for DQ and PCQ apparently provide different amount of information. Therefore, we consider DistQ with full budget. See our explanations in the first paragraph of Sec 5.2.
>
> (3) We next emphasize the effectiveness of DistQ. Given (2), it's hard to have a totally fair comparison among DistQ and baselines. The full and half budget settings actually provide a performance range of our method when compared with its baselines. Therefore, it's normal that DistQ (half) can't beat all its rivals in Fig 3. The fact that DistQ (half) can match or outperform baselines demonstrates in our opinion that it is possible to achieve good performance while asking informative and easier to answer queries. In addition, we believe that answering one DQ is easier than two PCQs (especially when not controlling for their hardness) based on our proposition. Besides, the goal of DistQ is to balance query informativeness and user-friendliness rather than only one of the two. Thus, we need to look at both performance (Fig.3) and query easiness (Fig.5, where our method outperforms others).
>
> (4) As for the suggested setting DistQ (half-half), we think this is identical to DistQ in our paper from the view of query quantification.
>
> 2. About easiness measurement:
>
> We consider the "predicted feedback" in terms of human side since we hope to quantify easiness by whether the human can provide correct feedback to queries. Similar query easiness measurement to our "wrongly predicted feedback" has also been used in previous work (**see the 2nd paper in the references list**). For the ground truth preference entropy of queries, it seems not very straightforward to see whether queries are easy to answer for humans compared with wrongly predicted number of feedback. But indeed, we acknowledge that this may also be a good measurement for easiness. We will use this entropy as a supplementary easiness measurement in our final version.
>
> 3. About essential references:
>
> Thank you for suggesting these recent works. We will discuss how our paper is related to them in our final version.
>
> 4. About user study:
>
> (1) For each method, 150 queries are answered. We conduct 10 rounds of evaluation after one training (line 402-406). During one training, the human answers 150 queries.
>
> (2) The authors of PEBBLE conducted similar (but easier) user study on Quadruped agent, where 200 PCQs are needed. Besides, they only claimed that the agent could succeed but didn't report the success rate. PEBBLE fails to work in our experiments, because the setting is harder and with a more limited query budget.

---

### Decision · Program_Chairs · 2025-05-01

**Decision:**

Accept (poster)

**Comment:**

This work introduces a novel distinguishability query to improve RLHF for robotics tasks, allowing human annotators to express preference strength by comparing two pairs of trajectories. Labelers first indicate which pair is easier to compare, then provide preference feedback only on the easier pair. This approach captures both preference strength and ordinal information, improving the efficiency of learning reward functions. Experiments show that DistQ is more data-efficient and user-friendly than traditional methods (reducing the cognitive load on the labeler).

The reviewers agree that the proposed method, experiment design, user study, analyses make sense and evaluation criteria are sound. The paper is well-motivated and well-written, addressing a relevant research question.

The authors present new experiments on harder tasks, videos of learned policies, details of the user study, and explained some misunderstandings in the discussion section during the discussion period. The additional experimental results and demonstrations presented enhance the quality of the paper and these should be included in the finalized version. The paper will also greatly benefit from adding more discussion about the cost of distinguishability queries compared to normal RLHF queries (and why a totally fair comparison  among DistQ and baselines is hard --- as discussed in responses to reviewer concerns).